# Machine Learning Techniques for Predicting the Energy Consumption/Production and Its Uncertainties Driven by Meteorological Observations and Forecasts

**Konrad Bogner [1,]*** [iD], **Florian Pappenberger [2]** [iD] **and Massimiliano Zappa [1]** [iD]

[1]  Swiss Federal Institute for Forest, Snow and Landscape Research WSL, Zürcherstrasse 111, 8903 Birmensdorf, Switzerland; massimiliano.zappa@wsl.ch

[2]  European Centre for Medium-Range Weather Forecasts ECMWF, Shinfield Park, Reading RG2 9AX, UK; florian.pappenberger@ecmwf.int

*  Correspondence: konrad.bogner@wsl.ch; Tel.: +41-44-739-2495

**Abstract:** Reliable predictions of the energy consumption and production is important information for the management and integration of renewable energy sources. Several different Machine Learning (ML) methodologies have been tested for predicting the energy consumption/production based on the information of hydro-meteorological data. The methods analysed include Multivariate Adaptive Regression Splines (MARS) and various Quantile Regression (QR) models like Quantile Random Forest (QRF) and Gradient Boosting Machines (GBM). Additionally, a Nonhomogeneous Gaussian Regression (NGR) approach has been tested for combining and calibrating monthly ML based forecasts driven by ensemble weather forecasts. The novelty and main focus of this study is the comparison of the capability of ML methods for producing reliable predictive uncertainties and the application of monthly weather forecasts. Different skill scores have been used to verify the predictions and their uncertainties and first results for combining the ML methods applying the NGR approach and coupling the predictions with monthly ensemble weather forecasts are shown for the southern Switzerland (Canton of Ticino). These results highlight the possibilities of improvements using ML methods and the importance of optimally combining different ML methods for achieving more accurate estimates of future energy consumptions and productions with sharper prediction uncertainty estimates (i.e., narrower prediction intervals).

**Keywords:** machine learning; monthly forecasts; predictive uncertainty

## 1. Introduction

The study of this paper has been inspired by the idea of analysing the potential benefits for managing water reservoirs and hydro-power plants by providing predictions of short- to long-term future energy consumptions in a region or a localized area. It is assumed that the energy consumption is mainly driven by meteorological conditions, for which forecasts will be available with different temporal and spatial resolutions. Under regular conditions, the production will be mostly driven by the energy price, but the limitations of the production will be given by hydro-meteorological boundary conditions. Thus, forecasts of the consumption and the production based on hydro-meteorological information could be gainful for the managers of the electricity grids and the power plants and providers of energy as well. If it is possible to identify statistically dependency structures between meteorological variables and the energy consumption/production, a regression model can be framed, which allows the prediction of the consumption/production. Such relationships could be modelled

with simple linear regression approaches (see for example [1,2], there is a great potential in improving the prediction models taking the nonlinearities and non-stationarities into account, which will bias the predictive skill and the forecast quality quite significantly at certain times during the weeks and months.

In this study, machine learning models have been tested in order to allow higher degrees of model complexity and to identify and to model dependency structures, which are hidden and therefore are most probably disregarded with simpler approaches. The different machine learning techniques applied are:

- Multivariate Adaptive Regression Splines (MARS): MARS build linear relationships between predictors and a target (predictand) by segmenting predictor variables. Possible nonlinear relationships can be identified by integrating all segments [3].
- Quantile Regression Neural Network (QRNN): Quantile Regression (QR) models [4] have been enhanced by Cannnon [5] using Neural Networks in order to incorporate nonlinearities.
- Kernel Quantile Regression (KQR): Use of kernel functions (weighting functions) to model dependencies non-parametrically, which allows modelling of both Gaussian and non-Gaussian data [6]. KQR is closely related to Support Vector Machines, but with different loss functions.
- Quantile Regression Forest (QRF): Based on decision tree models, a random forest is a tree-based algorithm, which builds several trees and combines their output by averaging each tree leaf in the forest, which helps to improve the generalization ability of the model. In quantile regression forests, all outcomes are stored, thus the quantiles from each tree leaf can be be calculated [7].
- Gradient Boosting Model (GBM): Also for the GBM, a decision tree model is chosen typically as a base model; however, ensembles of such prediction models are generated. The final GBM model is built iteratively by optimizing an arbitrary differentiable loss function [8].

The rationale behind choosing these techniques among the large number of ML methods available is primarily driven by the possibility of estimating quantiles directly and by the interpretability of results. Other Deep Learning methods have been omitted because they require large sample sizes [9,10].

These ML methods are compared with two linear models, the classical Multivariate Linear Regression (MLR) and the QR approach. Additionally, the different model outputs are optimally combined using the Nonhomogeneous Gaussian Regression (NGR) approach in order to calibrate the (monthly) forecasts [11]. Although there are studies available comparing different energy forecast models (e.g., [12]), only a few works take the predictive uncertainty into consideration (e.g., [13]). Thus, the main focus of this paper will be the derivation of quantiles of the predictions as a measure of uncertainty and how to verify the resulting quantile forecasts.

The coupling of energy models with numerical weather forecast models has been propagated by Taylor and Buizza [14], but, by the best knowledge of the authors, no monthly ensemble prediction system has been applied so far. This will be done the first time in this paper for a region of Switzerland. In order to calibrate the monthly forecasts, the ML based predictions will be optimally combined using the Nonhomogeneous Gaussian Regression method, which removes systematic bias of the forecasts and thus further improves the forecast skill.

The emphasis and novelty of this study is the derivation of predictive uncertainties based on ML approaches, which has been neglected in most analysis so far. Furthermore, there is a lack of in-depth studies regarding the usage of ensembles of monthly weather forecasts for the estimation of future energy consumptions and productions and its operational application. This study will be a first step to close the gap between research and application. An additional novelty is the analysis of possible improvements achieved through calibration, i.e., optimally combining different ML methods applied to monthly forecasts of the energy consumption/production.

The rest of the paper is structured as follows: Section 2 contains the detailed description of the case study and the methods. The results of the predictions and monthly forecasts will be presented and discussed in Section 3. Finally, some outlook will be given after the conclusion in Section 4.

## 2. Materials and Methods

For the ease of clarification, the main topics of the proposed research analysis, the structure of the study and the main steps of the modelling and evaluation chain are shown in Figure 1.

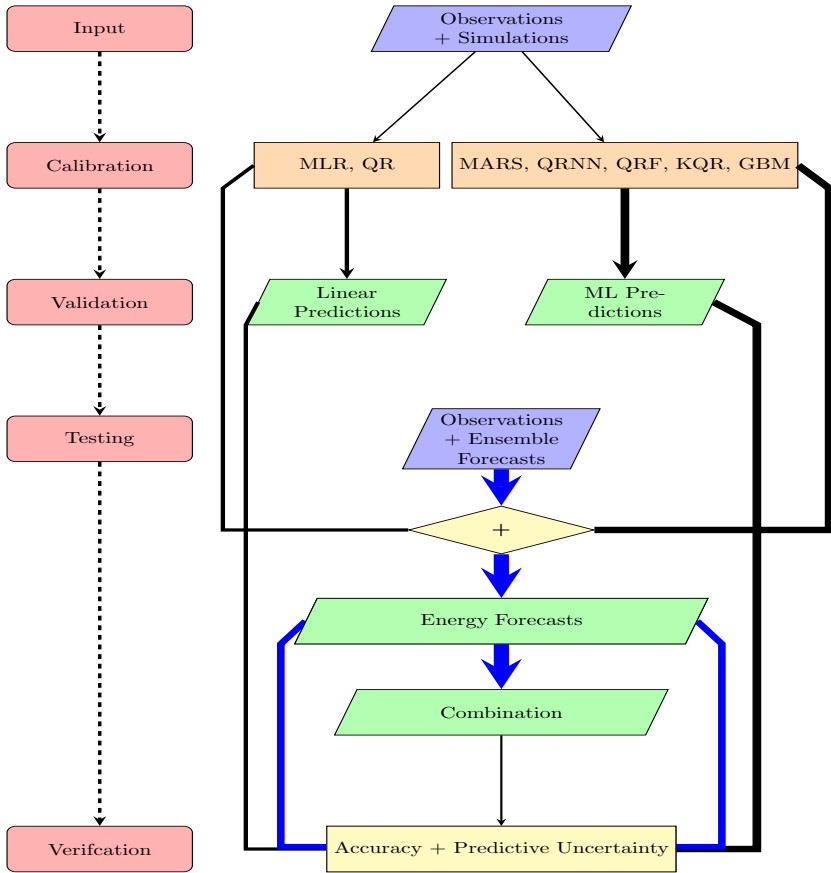

**Figure 1.** Flowchart of the modelling and evaluation chain. On the right side, the general structure of the study is shown. The left side shows the main topics of each step: starting from preparing the input data at the top, followed by the calibration and validation of the linear and Machine Learning (ML). models. In the testing phase, forecasts will be produced using the calibrated models and hydro-meteorological ensemble forecasts. After combining the different model outcomes (7 models × 51 ensemble members), the results of the predictions and the forecasts of the energy consumption/production will be verified with the target of quantifying the predictive uncertainty. The different thicknesses of the black and blue arrows indicate the different number of the used models and the derived predictions/forecasts.

The study starts with preparing the input data, which comprises observed hydro-meteorological observations and simulations and measurements of the energy consumption/production (see Section 2.1). These data are used for calibrating the different models under consideration (Section 2.2) and to make predictions of the energy consumptions/productions and to validate the predictive quality and uncertainties (Section 2.5). In the following testing phase, these calibrated models will be applied using ensemble hydro-meteorological forecast data consisting of 51 members as input in order to produce daily energy consumption/production forecasts for the upcoming months. Additionally, these suchlike created ensembles for each model will be aggregated to one optimally combined model (Section 2.4). Finally, all these outputs from the predictions and the forecasts (with and without combination) will be analysed and some verification measurements will be calculated with the final target of comparing the various models and to highlight the possibilities for estimating

reliable uncertainty ranges. All the different steps of this study will be explained next, starting with a description of the input data and a brief summary of the different ML techniques.

### 2.1. Data

The analysis is based on the energy consumption and production data available from www.swissgrid.ch. The data are provided spatially aggregated for each canton in Switzerland with 15 min resolution starting from the year 2015 and are updated regularly with a delay of one month. For this study, the data from 1 January 2015 to 31 October 2018 have been used. Since the ultimate goal is the testing of the predictability of monthly forecasts, these energy data are aggregated to daily sums, which corresponds to the resolution of the meteorological monthly forecast data available in this study. As input only, those meteorological variables from MeteoSwiss have been selected, for which information is available in the calibration period, in real-time and as a monthly forecast. Additionally, the factors Weekday and Holiday have been included (similar to [15]). In Figure 2, the location of the study area of the canton Ticino is shown in blue and the Verzasca catchment is highlighted in red. Based on previous studies of applicability of monthly weather forecasts for the optimization of the hydro power plant production (see [16]), the meteorological data spatially aggregated for the Verzasca catchment have been used as a surrogate for the canton Ticino (Figure 3) and the runoff, production and consumption data are for the Canton Ticino (see Figure 4).

It should be stressed that this study is based on simplified assumptions in that the meteorological conditions of the Verzasca catchment are representative for the whole Canton Ticino, which represents about 10% of the area of the Canton Ticino. However, several important hydro power plants, the main drivers for the energy production in Ticino, are located also outside the Verzasca catchment and thus the runoff from the whole Canton is taken as a surrogate variable. Unfortunately, the meteorological observations and forecasts have been available for the Verzasca catchment only. Missing or erroneous values have been replaced by applying a smoothing spline interpolation method.

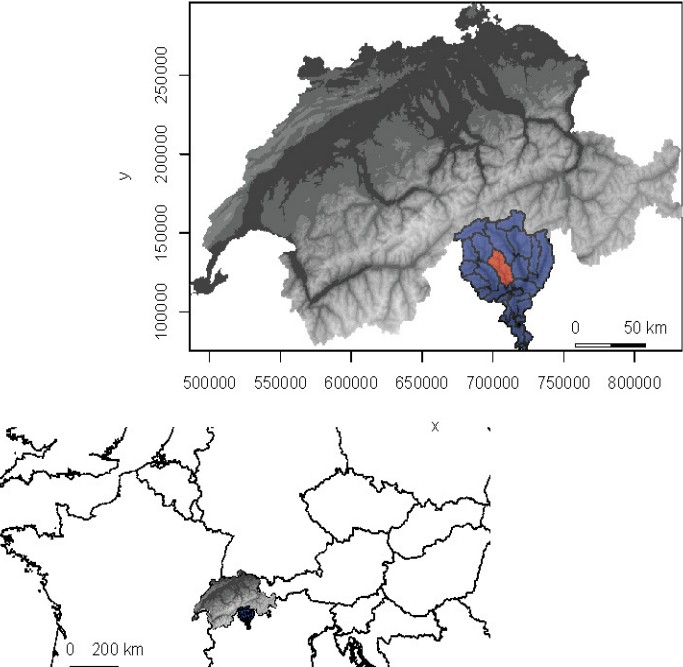

**Figure 2.** Location of the Canton Ticino (blue) in Central Europe (lower part) and in Switzerland (upper part) with the Verzasca catchment shown in red.

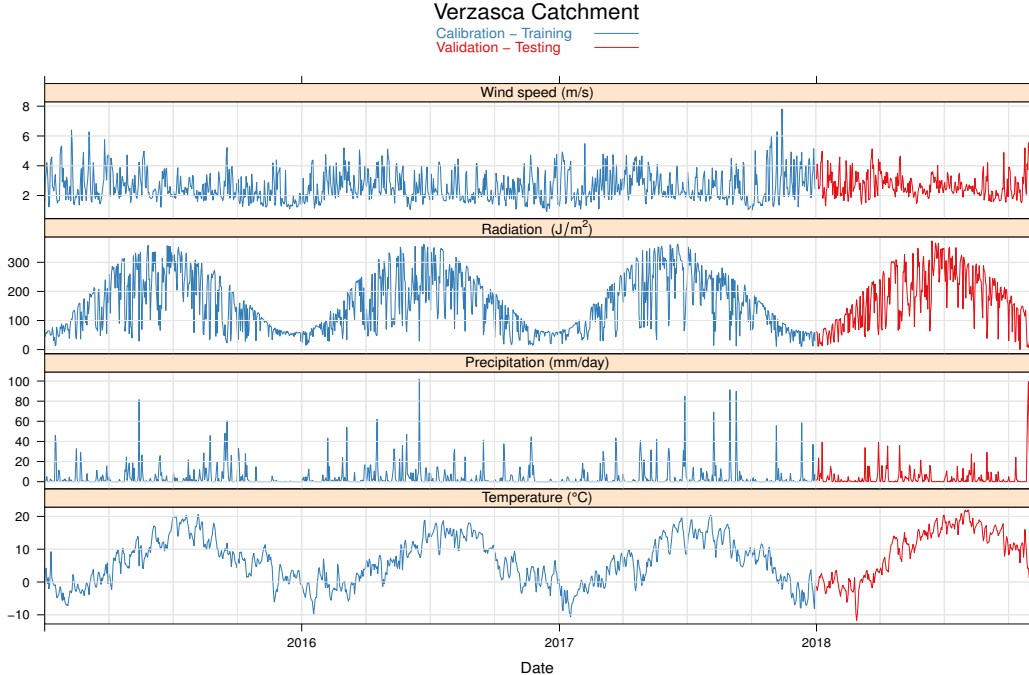

**Figure 3.** Daily aggregates from the Verzasca catchment of the meteorological input data: Wind, Radiation, Precipitation, Temperature. In blue, the calibration (training) period (1 January 2015–31 December 2017) and in red the validation (testing) period (1 January 2018–31 October 2018) is shown.

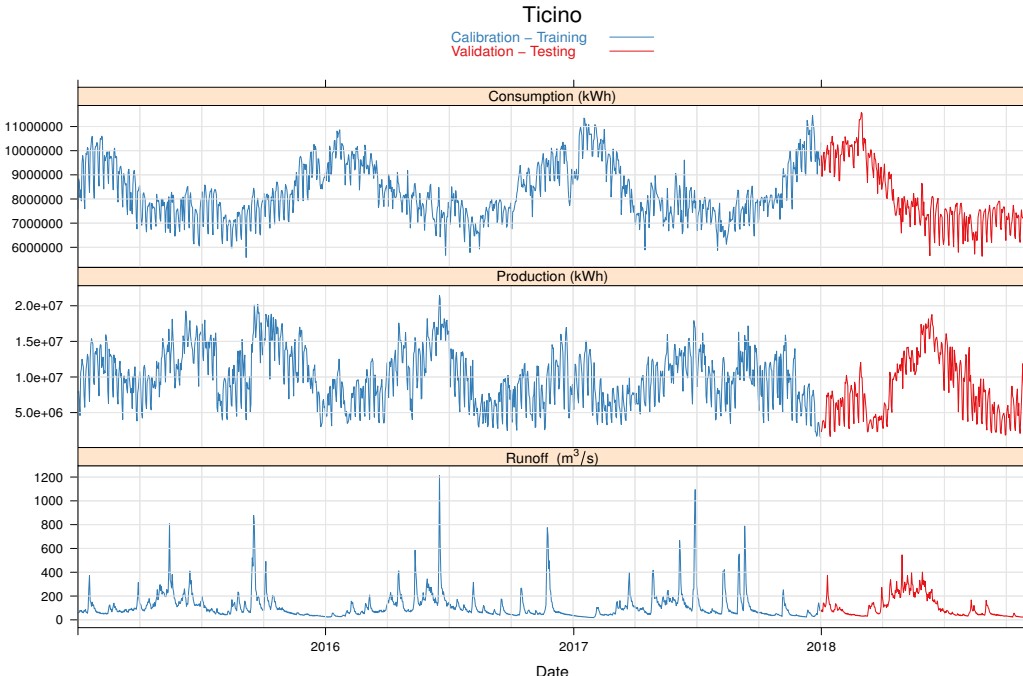

**Figure 4.** Daily aggregates from the Canton Ticino of the consumption, production and surface runoff data. In blue, the calibration (training) period (1 January 2015–31 December 2017) and in red the validation (testing) period (1 January 2018–31 October 2018) is shown.

In Table 1, the used data are summarized. It is important to note that the meteorological data have been interpolated to a grid with 200 m resolution for the Verzasca catchment. In addition, although the proxy for the surface runoff is simulated for the whole Canton, the spatial resolution of the hydrological model is only 500 m and thus prone to huge uncertainties and errors as well. Thus, both aspects,

the uncertainty caused by downscaling and interpolation methods and the small fraction taken as a meteorological surrogate of the total area, should be taken into consideration in the verification results. Most probably, the analysed methods could yield better results with more precise input data with higher spatial resolution. Thus, one additional interesting aspect of this analysis is the problem of upscaling and how this very localized information of the Verzasca catchment can be used to predict the energy production/consumption for a whole canton.

**Table 1.** Categorical and hydro-meteorological data [units] used for modelling the consumption/ production of the Canton Ticino.

| Dependent | Weekday | Holiday | Temp. | Precip. | Radiation | Wind | Runoff |
|---|---|---|---|---|---|---|---|
| Consumption [kWh] | 1–7 | 0–1 | [°C] | [mm] | [J/m$^2$] | [m/s] | |
| Production [kWh] | 1–7 | 0–1 | [°C] | [mm] | [J/m$^2$] | [m/s] | [m$^3$/s] |

In order to make the results of the different models comparable, the input data have been pre-processed equally. In a first step, all the variables have been standardized (i.e., subtraction of the mean and divided by the standard deviation), which is also known as feature scaling in ML. Preliminary analysis of the different input variables identified the supreme importance of the Temperature for the consumption model and the Runoff for the production model. In Figure 5, the results of the relative influence of variables analysis for the GBM (see [17]) for the consumption and production model are shown. The other methods show similar results for the ranking of the importance of the different variables.

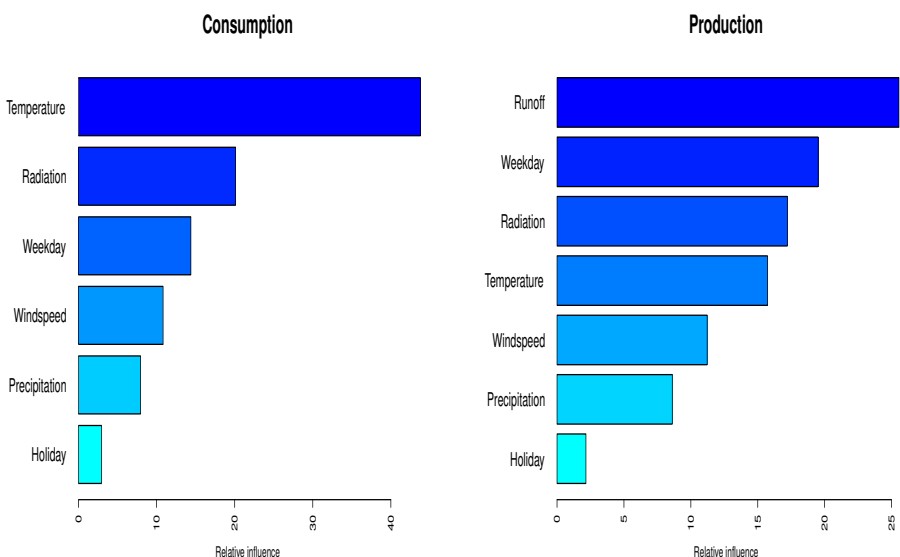

**Figure 5.** Relative influences (in percentages) of the variables based on the Gradient Boosting Machine (GBM) for the consumption (**left**) and the production (**right**) model.

In a second step, the temperature (for the consumption models) and the runoff (for the production models) have been decomposed into different scales by the use of the non-decimated wavelet method (details can be found in [18]). For this study, the most simple Haar wavelet has been applied, which corresponds to, simply speaking, successive differencing and smoothing of the variable over steadily increasing time-intervals. This wavelet transformed series allow the capturing of different intrinsic details relevant for different time scales, which could be of importance for predicting scale dependent properties and which would hardly be identified otherwise. The advantages of decomposition methods for forecasting models have been shown in the work of Hu [19].

For analysing and comparing the ML methods, the data sets have been split into a training period (1 January 2015–31 December 2017) and a testing period (1 January 2018–31 October 2018) as indicated in Figures 3 and 4. In the first part of this analysis, observed meteorological data from the Verzasca catchment and surface runoff simulated by the hydrological model PREVAH [20] for the Canton of Ticino have been used in order to exclude uncertainties and errors stemming from numerical weather prediction systems. In the second part, the meteorological observations have been replaced with the monthly ensemble forecasts (ENS) calculated at the European Centre of Medium-range Weather Forecast (ECMWF) and delivered from MeteoSwiss. The version of the meteorological forecasts used in this study consists of 51 ensemble members and are issued twice a week with lead-times of 45 days (see [21] for details). Within this study, only 32 days are used as current skill levels do suggest no utility for further forecast horizons.

### 2.2. Models

For the MLR model, the reader is referred to the statistical literature (see, for example, [22]). In order to allow maximal transparency and reproducibility, all these analysed techniques are implemented in the freely available statistical scripting language R [23]. However, only concise details about the used R packages and their parameter settings will be given here and the reader is referred to the corresponding web pages for more details. Some of these ML techniques are also included in the R package **caret** [24], which facilitates a fine-tuning and sensitivity analysis of the model hyper parameters.

#### 2.2.1. Multivariate Adaptive Regression Splines (MARS)

MARS is a nonparametric statistical method in which the data sets are partitioned into basis functions (BFs), which represent piecewise linear segments (splines) of differing gradients. The connection/interface points between the pieces, called knots, are placed at random positions within the range of each input variable. The MARS estimate of the unknown regression function $f(x)$ can be written as an additive function of the product basis functions [25]:

$$f(x) = \beta_0 + \sum_{m=1}^{M} \beta_m B_m(x), \tag{1}$$

where $\beta_0$ is the coefficient of the constant basis function $B_0(x) = 1$, $B_m(x)$ is the $m$th basis function, which may be a single spline function or product of two or more, $\beta_m$ is the coefficient of the basis function, and $M$ is the number of basis functions in the model. The essential part of the MARS model is the so-called hinge function, which maps a variable $x$ to $x^\star$ as:

$$x^\star = max(x - c, 0), \tag{2}$$

where $c$ is the knot of the basis function. The basis function $B_m(x)$ itself is defined as [26]:

$$B_m(x) = \prod_{k=1}^{K_m} \left[ max \left( s_{k,m} \left( x_{v(k,m)} - t_{k,m} \right), 0 \right) \right], \tag{3}$$

where $v(k, m)$ represents the explanatory variables associated with the basis function $B_m$ and $K_m$ is the level of interaction between $v(k, m)$ variables and $t_{k,m}$ is the location of the knot. The flexibility of the MARS model results from the smooth connection of splines without making assumptions about the functional relationships between the dependent variable and the predictors. The MARS algorithm is based on an adaptive regression approach running a forward and a backward procedure for generating the base functions and selecting the locations of the knots. At each forward step, the entire domain is split into subregions and the knots and their corresponding basis functions are added; at the backward step, the redundant basis functions are deleted to avoid overfitting [27]. This

process is known as "pruning" and the optimal number of knots can be find using cross-validation. After determination of the optimal MARS model, the parameter relative importance based on the contributions from the input variables can be assessed with the help of the analysis of variance (ANOVA) decomposition procedure [28]). In order to estimate the quantiles of the predicted variable, a second model will be fitted to the residuals. In the most simple approach, a linear variance model is applied, which incorporates heteroscedasticity properties of the residuals (i.e., linear changes in the spread of the residuals).

MARS is implemented in the R package **earth** [29] and the model parameters have been tuned with **caret**.

### 2.2.2. Quantile Regression (QR)

Standard linear regression models focus on finding a conditional mean function describing the linear relationship between the predictor and the independent variable(s), whereas Quantile Regression (QR) models look at different quantiles of the response defined by the conditional-quantile function [4,30,31]. Thus, the full conditional distributional properties of the response variable can be analysed without making any assumptions about the error distribution [16]. Hence, in QR models, the relationship between the predictor and the independent variable(s) will not be described with a single slope parameter, like in linear regression models, but a set of parameters $\beta_\tau$ dependent on the quantile $\tau$ have to be estimated. Koenker and Bassett [30] define the $\tau$th regression quantile $(0 < \tau < 1)$ as any solution, $\beta_\tau$, to the quantile regression minimization problem

$$\min_{\beta_\tau \in \mathbb{R}} \sum_{i=1}^{n} \rho_\tau \left( y_i - \xi_\tau \left( x_i, \beta_\tau \right) \right), \tag{4}$$

where $\rho_\tau \left( y_i - \xi \left( x_i, \beta_\tau \right) \right)$ is a function of $\tau$ and $y_i - \xi_\tau \left( x_i, \beta_\tau \right)$. This kind of loss function is most often called check or pinball loss function and is defined as [4,32]:

$$\rho_\tau \left( y_i - \xi \left( x_i, \beta_\tau \right) \right) = \begin{cases} \tau \left( y_i - \xi \left( x_i, \beta_\tau \right) \right) & \forall y_i \geq \xi_\tau \left( x_i, \beta_\tau \right) \\ \left( \tau - 1 \right) \left( y_i - \xi \left( x_i, \beta_\tau \right) \right) & \forall y_i < \xi_\tau \left( x_i, \beta_\tau \right), \end{cases} \tag{5}$$

where $\{ x_i : i = 1, ..., n \}$ denotes a sequence of explanatory variables and $\xi_\tau \left( x_i, \hat{\beta}_\tau \right)$ is formulated as a linear function of parameters. In Koenker [4], a description is given how the resulting minimization problem can be solved by linear programming methods.

### Quantile Regression Neural Network (QRNN)

The theoretical support for the use of quantile regression within an Artificial Neural Network (ANN) in order to estimate potentially nonlinear quantile models has been outlined by White [33] and in [5,34,35] some applications are shown.

In QR, the parameters $\beta_\tau$ have to be optimized by solving the optimization problem defined in Equation (4). In an ANN with a hidden layer, the parameters include the hidden layer weights, the hidden layer biases, the output weights and the output biases. In order to solve nonlinearities, an activation function (e.g., the sigmoid function) has to be applied to each node (neuron) in the hidden and the output layer, which maps the inner dot product of the weights and the input (plus bias) into a specified range (e.g., between 0 and 1 in case of the sigmoid function). The risk of overfitting the ANN can be reduced by adding a weight decay regularization term to the check function in Equation (4).

One major drawback of this approach is the separate estimations of the quantiles, which could lead to erroneous crossing of quantiles [36]. This problem has been solved by [37], who introduced an efficient, flexible nonlinear quantile regression model, the monotone composite quantile regression neural network (MCQRNN). This method estimates simultaneously multiple non-crossing quantile functions and allows for optional monotonicity, positivity/non-negativity, and additivity constraints. Therefore, it combines

elements drawn from the standard QRNN model [5,33,34], the monotone multi-layer perceptron (MMLP) [38,39] and the composite QRNN (CQRNN) [40]. The basis of the MCQRNN is the multi-layer perceptron (MLP) neural network with partial monotonicity constraints [41]. The MCQRNN model is available in **QRNN** [5,37]. The number of hidden layers have been optimized using a grid search approach by running the model with different numbers of hidden layers and choosing the number, which minimizes the validation error function (for example the Mean Absolute Error).

Kernel Quantile Regression (KQR)

Support Vector Machines (SVM) have been applied successfully to many different real-world problems like electricity load [42] and consumption forecasting [43] and is based on the statistical learning theory [44]. Basically, the SVM maps nonlinearly the original data into a higher dimensional feature space using a kernel (e.g., the Gaussian Radial Basis Function) before solving the machine learning task as a convex optimization problem. For Support Vector Regression (SVR) purposes, a loss function has to be trained, which penalizes high and low misestimates. Therefore, a flexible tube of minimal radius is formed symmetrically around the estimated function, the so-called hyperplane, such that the absolute values of errors less than a certain threshold $\epsilon$ are ignored both above and below the estimate. In this manner, points outside the tube, the support vectors, represents the hyperplane and are penalized. Thus, support vectors are the most influential instances that affect the shape of the tube and define the margins of these hyperplanes. Hence, a multiobjective function $f(x)$ is constructed from the loss function with at most $\epsilon$ deviation from the actually target values $y_i$ and as flat as possible properties of the tube. Flatness in case of a linear function $f(x) = \langle \omega, x \rangle + b$, where $\langle ., . \rangle$ denotes the dot product, $\omega$ the weights and $b$ the bias, can be achieved by minimizing the norm $\|\omega\|^2 = \langle \omega, \omega \rangle$. These constraints results in a convex optimization problem [45]):

$$\text{minimize} \quad \frac{1}{2}\|\omega\|^2 \tag{6}$$

$$\text{subject to} \begin{cases} y_i - \langle \omega, x_i \rangle - b \leq \epsilon \\ \langle \omega, x_i \rangle + b - y_i \leq \epsilon \end{cases} \tag{7}$$

and can be solved using quadratic programming techniques. For nonlinear functions, the data have to be mapped into a higher dimensional space, which can be done efficiently by the use of kernels. Most often, the Gaussian radial basis function is applied for SVR, which is a general purpose kernel and is defined as:

$$k(x, y) = \exp -\sigma\|x - y\|^2, \tag{8}$$

where $\sigma$ is the width parameter controlling the trade-off between error due to bias and variance [46]. In case of the KQR, the loss function given in Equation (5) will be used to estimate $f(x)$. More detail about Support Vector Regression and KQR can be found in [6,45].

The KQR is one out of many different kernel based ML techniques implemented in **kernlab** [47]. The parameter $C$, which regularizes the weight assigned to the loss function, i.e., the minimization of the error, and the geometric property, i.e., the flatness, of the tube, is tuned with a grid search approach.

Quantile Regression Forest (QRF)

Random Forest (RF) models have been explained in detail for example in [48]. Here, only a brief summary of this ML technique will be given. Regression trees divide the original data space into small partitions in which the interaction is easier to be fitted.The recursive partition makes the model look like a tree. Each node on the tree represents a small data space corresponding to a simple model [49]. The RF model creates a large number of trees as base models by randomly selecting a subset of attributes in each splitting on randomly selected subsets of the training data. Essentially, RF combines many regression trees into an ensemble to produce more accurate regressions by drawing several bootstrap samples from the original training data and fitting a tree to each sample [50]. Within a RF,

predicted responses $\hat{Y}$ for $k = 1, \ldots, m$ new data points resulting from a vector of predictors, $X_k$, are modelled as a weighted average of responses, $Y$ of $y = 1, \ldots,$ training data points, with weights $W$ depending on the predictors [51]

$$\hat{Y}_k = \sum_{j=1}^{n} W_j(X_k) Y_j. \tag{9}$$

The weights sum to one and are nonnegative. Each RF is composed of many trees. Each tree $t$ is grown by repeatedly splitting a bootstrap sample (an independent sample selected with replacement) of the training data. Each split $s$ represents a value of a predictor. The combination of predictor and split is derived such that it leads to the smallest total impurity, defined as the sum of squared deviations about the group mean. Splitting occurs repeatedly until a minimum number of observations in the partition is attained, at which point the partition becomes a terminal node. New predictions of the response are the average of all observed values that fall in each terminal node for each tree, averaged over all trees. Averaging over all trees provides predictions that are dependent on the full training data set including responses and predictors. Random forests give an accurate approximation of the conditional mean of a response variable. However, a generalisation of random forests introduced by [7] allows the estimation of conditional quantiles. This method is called Quantile Regression Forests. The trees in QRF are growing as in the standard random forests algorithm. The conditional distribution is then estimated by the weighted distribution of observed response variables, where the weights attached to observations are identical to the original random forests algorithm. The main difference between QRF and RF is that, in the latter, for each node in each tree, only the mean of the observations that fall into this node are kept and all other information are neglected. In contrast, quantile regression forests keep the value of all observations in this node, not just their mean, and assesses the conditional distribution based on this information. In [52], QRF has been applied successfully for calibrating meteorological ensemble forecasts.

QRF is implemented in the R package **quantregForest** [53]. The tuning of the model parameters (e.g., **mtry**: the number of variables randomly sampled as candidates at each split) has been done with the greed search approach.

Gradient Boosting Machine (GBM)

As mentioned previously, common ensemble techniques like RF rely on simple averaging of models in the ensemble. The boosting methods differ by constructing an ensemble sequentially by training, at each particular iteration, a new weak, base-learner model, most frequently a decision tree, with respect to the error of the whole ensemble learnt so far. An overview of boosting techniques is given by [54].

In [17,55], a gradient-descent based formulation of boosting methods was derived, where the new base-learners are constructed to be maximally correlated with the negative gradient of the loss function, associated with the whole ensemble. Given a dataset $(x, y)_{i=1}^{N}$, where $x = (x_1, \ldots, x_d)$ refers to the explanatory input variables and $y$ to the corresponding labels of the response variable, the objective is to reconstruct the unknown functional dependence $x \xrightarrow{f} y$ with our estimate $\hat{f}(x)$, such that some specified loss function $\psi(y, f)$ is minimized:

$$\hat{f}(x) = \arg\min_{f(x)} \psi(y, f(x)). \tag{10}$$

After initialization of $\hat{f}_x$ to be a constant the Gradient Boost algorithm of Friedmann [17] proceeds with the iteration of

- Computation of the negative gradient:

$$z_i = -\frac{\partial \psi(y_i, f(x_i))}{\partial f(x_i)} \bigg|_{f(x_i) = \hat{f}(x_i)}. \tag{11}$$

- Fitting a regression model, $g(x)$, predicting $z_i$ from the covariates $x_i$
- Choosing a gradient descent step size as:

$$\rho = \arg\min_{\rho} \sum_{i=1}^{N} \psi(y_i, \hat{f}_i + \rho g(x_i)). \tag{12}$$

- Updating the estimate of $f(x)$ as

$$\hat{f}(x) \leftarrow \hat{f}(x) + \rho g(x). \tag{13}$$

At each iteration, the algorithm determines the direction, the gradient in which it needs to improve the fit to the data and selects a particular model.

For the squared error cost function, for example, the negative gradient is simply equal to the residuals and thus the trees are iteratively fitted to the residuals [56]. However, gradient boosting can be applied to any differentiable cost function. Ref. [57] showed that the gradient tree boosting algorithm predicts the quantile, when a weighted cost function is implemented.

The performance of gradient boosting can be improved with bagging [17]. For example, in decision tree models, each tree is grown using a simulated training dataset, constructed from a fraction of the original training set using bootstrapping. The problem of overfitting can be avoided if the contribution of each tree is scaled by a learning rate (shrinkage). This helps to constrain the fitting procedure and thus balance the predictive performance of the resulting model (e.g., [58]).

In this work, GBM has been applied with the help of the R package **gbm**. The number and depth of trees, the learning rate, and the bagging percentage are hyper-parameters that must be tuned. The size and number of trees are selected using cross-validation.

### 2.3. Estimation of the Quantiles

For the Quantile Regression based models (QR, QRNN, QRF, KQR, GBM), the quantiles can be estimated directly by implementing the check function (Equation (5)) as a loss function. For the MLR and the MARS model, the quantiles can be estimated from prediction intervals for specified confidence levels, whereas, for the MLR, the model error $\epsilon$ is assumed to be normally distributed with zero mean and constant variance, and the residuals of the MARS model can be modelled with an extra variance model in order to allow heteroscedasticity. In the most simple case, the residuals are assumed to vary linearly with the predicted response. In this study, possibly nonlinearities in the variance model have also been investigated, but did not show any improvements.

### 2.4. Forecast Combination

A possibility to address under-dispersion and forecast bias is the use of the Non-homogeneous Gaussian Regression (NGR) method or Ensemble Model Output Statistics (E-MOS) and is based on multiple linear regression. More information about the MOS technique can be found for example in [59,60]. Its extension for ensembles is explained in [11] and a brief summary of this method is given hereafter. Let $y$ denote again the variable of interest (e.g., energy consumption) and let $f_1, f_2, \ldots, f_K$ be the corresponding forecasts of the $K$ ML models. If $\mathcal{N}(\mu, \sigma^2)$ denotes a Gaussian density with mean $\mu$ and variance $\sigma^2$, the NGR predictive distribution is given by Gneiting, et al. [11]:

$$y | f_1, \ldots, f_K \sim \mathcal{N}(a_0 + a_1 f_1 + \cdots + a_K f_K, b_0 + b_1 s^2), \tag{14}$$

$$\text{where } s^2 = \frac{1}{K} \sum_{k=1}^{K} \left( f_k - \frac{1}{K} \sum_{k=1}^{K} f_k \right)^2.$$

It should be noted that each ML model is driven by a hydro-meteorological forecast ensemble consisting of $N$ members. However, these ensemble members are exchangeable, i.e., at each forecast initiation, the members are chosen randomly and have no individually distinguishable characteristics.

Thus, the predictive mean is equal to the regression estimates for the ensemble means and forms a bias-corrected weighted average of the different forecasts (*K* ML models), whereas the predictive variance depends linearly on the variance of the *K* ML models plus the variance of *N* ensemble members of each ML model.

The coefficients $a_0, \ldots, a_k, b_0$, and $b_1$ are estimated by the maximum likelihood method, which can be done by maximising the log-likelihood function of the model (Equation (14)) and is equivalent to minimising the ignorance score (see [11] for further details).

## 2.5. Verification

In most publications, the ML methods are evaluated using some classical verification measures, like the Mean Absolute Error (MAE), Mean Absolute Percentage Error (MAPE). In this paper, the Coefficient of Determination will be used, which is also called $R^2$ in statistics and is the proportion of the variance in the dependent variable that is predictable from the independent variable(s):

$$R^2 = 1 - \frac{\sum(y_i - \hat{y}_i)^2}{\sum(y_i - \bar{y})^2}, \tag{15}$$

where $y_i$ are the observed values, $\hat{y}_i$ are the predicted values, and $\bar{y}$ is the mean of the observations (see, for example, [61]). For the monthly forecasts, the Nash–Sutcliffe efficiency coefficient has been used, which is almost identical to the $R^2$ (see [62]). The difference is that the predicted values $\hat{y}_i$ of the statistical models in Equation (15) are replaced with model simulations, which are not directly inferred from the observed values and thus allows negative results (whereas the $R^2$ is defined between zero and one only).

All these measures are useful for verifying the accuracy of single, deterministic forecasts. Since the results in this study will include informations about the predictive uncertainty, derived from quantiles, such measures will only highlight some aspects of the forecast performance. Thus, the prediction and forecast quality has to be evaluated not only regarding accuracy, but should also include the sharpness of the forecast. This means that the verifying observation should not only be as close as possible to the forecast mean (median), but also the prediction intervals should be as narrow as possible. The assessment of the forecast accuracy and sharpness can be verified by the Continuous Ranked Probability Score (CRPS, [63]). Although this measure is usually applied to continuous variables, it can be re-written in order be applied to quantiles directly (see [64] for details). The CRPS compares the forecast probability distribution with the observation and both are represented as cdfs. If *F* is the predictive cdf and *y* is the verifying observation, Ref. [64] showed that the CRPS can be defined equivalently as standard form,

$$CRPS(F, y) \;\; = \;\; \int_{-\infty}^{\infty} (F(t) - I\{y \le t\})^2 \, dt, \quad \text{and as} \tag{16}$$

$$= \;\; 2\int_0^1 \left( I\left\{ y < F^{-1}(\tau) \right\} - \tau \right) \left( F^{-1}(\tau) - y \right) d\tau. \tag{17}$$

Thus, in the standard form (Equation (16)), an ensemble of predictions can be converted into a piecewise constant cdf with jumps at the different models (ensemble members), and $I\{.\}$ is a Heaviside step function, with a single step from 0 to 1 at the observed value of the variable. For the quantile forecast $q_\tau = F^{-1}(\tau)$, the integrand in Equation (17) equals the Quantile Score (QS), i.e., the mean of the check function (Equation (5)). More details about the QS can be found in [28,65]. This means that the CRPS corresponds to the integral of the QS over all thresholds, or likewise the integral of the QS over all probability levels [64,66]. Hence, the CRPS averages over the complete range of forecast thresholds and probability levels and is negatively oriented, meaning the smaller the better. It is also possible to construct weighted versions of the CRPS emphasizing user defined regions of interest (see [67]). Following the work of [64], four different quantile weight functions have been

analysed with center, tails, right and left tail emphasis. In Table 2, these functions are summarized. This decomposed analysis could be interesting for applications, where not only the quality of the forecast is analysed during average conditions, but also to evaluate which model is better for predicting extremes (for example during dry and hot periods, resp. cold and wet periods).

**Table 2.** Weight functions for the quantile weighted versions of the CRPS, where $q$ is the quantile forecast, defined in [67].

| Emphasis | Quantile Weight Function |
|---|---|
| w1: center | $q(1-q)$ |
| w2: tails | $(2q-1)^2$ |
| w3: right tail | $q^2$ |
| w4: left tail | $(1-q)^2$ |

## 3. Results and Discussion

Regarding the evaluation and comparison of the predictive uncertainties of various ML models, only a very limited number of works exists, mostly with the focus on one specific model only. For example, in [68], confidence intervals for the Gaussian Process (GP) regression have been reported and, in [69], prediction intervals have been studied for the energy load forecasts using the generalized extreme learning machine.

Most recently, a comprehensive literature review article about the application of ML methods for the prediction of energy consumption has been published by Mosavi and Bahmani [70] showing that the number of publications dramatically increased within the last 5 to 10 years. However, the majority of the papers are dealing with short-term (hourly to daily) and long-term (yearly) predictions and only a few (e.g., [68]) with medium-term predictions (weekly to monthly time scales). The main difference to the results presented here is that non of these studies used monthly forecasts with daily resolution, but monthly aggregates (sums, averages) where the information of the temporal evolution within the upcoming weeks got lost. The same is true for the prediction of the energy production (see [71]). In [72], different models for forecasting the wind power generation have been compared with different time horizons, but without using hydro-meteorological forecasts.

Additionally, no articles have been found from a keyword search including CRPS (Continuous Ranked Probability Score) AND Machine Learning AND energy consumption OR production. In addition, the method of the NGR approach for optimally combing the different forecasts has not been applied for predicting the energy consumption/production.

### 3.1. Evaluation Based on Observed Meteorological Input Data

The different models for the prediction of the consumption and the production have been evaluated using the Coefficient of Determination ($R^2$) and the Continuous Ranked Probability Score (CRPS). Whereas the former has been calculated using the pairs of measured data and the median (0.5 quantile) of the different ML methods, the latter has been evaluated for the $\{0.01, 0.25, 0.5, 0.75, 0.99\}$ quantiles.

The effect of applying the wavelet transformation to the most important variables (i.e., the temperature for the consumption and the runoff for the production model) has been analysed at first. The results for the consumption model without wavelet transformation are shown in Table 3 in brackets indicating that the $R^2$ of all the models improved after including the wavelet decomposed variables. The same order of improvements could be achieved for the production model (not shown here). For the training of the different ML models, the cross validation methods have been applied by repeated randomly splitting the training period into a calibration and a validation period (with a minimum block size of one month to circumvent errors caused by distorted autocorrelation properties). In order to apply the modified CRPS (see Section 2.5, Equation (17)), the quantiles of the predictions have to be calculated as explained in Section 2.3.

The results of the $R^2$ and the CRPS and its adaptation for emphasising four different quality criteria (see Table 2) for the consumption and the production models are given in Tables 3–6.

**Table 3.** $R^2$ for predicting the consumption for the training and the testing period in the Canton Ticino. In brackets, the results for applying the different models without using the wavelet transformed temperature variable and in bold the results of the best performing models are shown.

| Consumption | MLR | MARS | QR | QRNN | QRF | KQR | GBM |
|---|---|---|---|---|---|---|---|
| Training | 0.68 | 0.84 | 0.69 | 0.87 | 0.82 | 0.89 | 0.89 |
| | (0.62) | (0.76) | (0.62) | (0.76) | (0.75) | (0.82) | (0.78) |
| Testing | 0.72 | 0.82 | 0.73 | 0.80 | **0.83** | 0.81 | **0.83** |
| | (0.66) | (0.74) | (0.66) | (0.73) | (0.77) | (0.68) | (0.71) |

**Table 4.** CRPS for the testing period of the predicted quantiles of the consumption in the Canton Ticino (in bold the results of the best performing models are shown).

| Consumption | MLR | MARS | QR | QRNN | QRF | KQR | GBM |
|---|---|---|---|---|---|---|---|
| CRPS | 3.75 | 2.97 | 3.88 | 3.29 | **2.88** | 3.13 | 2.93 |
| w1 (center) | 0.77 | 0.61 | 0.79 | 0.65 | 0.60 | 0.64 | **0.59** |
| w2 (tails) | 0.64 | 0.54 | 0.71 | 0.67 | **0.49** | 0.58 | 0.56 |
| w3 (right tail) | 1.05 | 0.87 | 1.16 | 0.99 | **0.81** | 0.96 | 0.92 |
| w4 (left tails) | 1.14 | 0.89 | 1.14 | 0.98 | 0.88 | 0.89 | **0.83** |

**Table 5.** $R^2$ for predicting the production for the training and the testing period in the Canton Ticino.

| Production | MLR | MARS | QR | QRNN | QRF | KQR | GBM |
|---|---|---|---|---|---|---|---|
| Training | 0.45 | 0.62 | 0.44 | 0.73 | 0.72 | 0.70 | 0.75 |
| Testing | 0.45 | **0.61** | 0.43 | 0.51 | 0.54 | 0.59 | **0.61** |

**Table 6.** CRPS for the testing period of the predicted quantiles of the production in the Canton Ticino (in bold the results of the best performing models are shown).

| Production | MLR | MARS | QR | QRNN | QRF | KQR | GBM |
|---|---|---|---|---|---|---|---|
| CRPS | 16.92 | **13.83** | 17.16 | 15.86 | 16.06 | 15.10 | 15.02 |
| w1 (center) | 3.50 | **2.85** | 3.55 | 3.13 | 3.32 | 3.09 | 3.06 |
| w2 (tails) | 2.91 | **2.42** | 2.96 | 3.34 | 2.77 | 2.72 | 2.7 |
| w3 (right tail) | 5.01 | **4.04** | 5.14 | 4.46 | 4.68 | 4.36 | 4.21 |
| w4 (left tail) | 4.91 | **4.08** | 4.92 | 5.14 | 4.74 | 4.55 | 4.69 |

All of the ML methods show an improvement of the accuracy for the daily predictions of the consumption of about 10% in the testing period. It is interesting to see that there is no clear preference of the ML method and all work almost equally well ($R^2$ between 0.80 and 0.83). Regarding the production, the MARS, KQR and GBM method show better results in comparison to QRNN and QRF. However, also for the production, all of the ML methods show an improvement of the $R^2$ between 5–15%.

Another important aspect is the analysis of the predictive uncertainty. In addition, regarding this quality, evaluated with the CRPS, the ML methods are able to improve the skill. However, there is a greater variability between the different methods. For the consumption models, the QRF method shows the best sharpness and the reliability properties indicated by the lowest CRPS value. When different weights to the quantiles are applied (as defined in Table 2), the QRF is better in representing the tails and the right tail of the variables range, whereas the GBM method is preferable, when the emphasis is on lower tails. Both methods are comparable good in improving the quality, if the emphasising region is in the center. For the production model, the MARS model shows the best results for the CRPS and the weighted CRPS.

### 3.2. Monthly Forecasts

In order to evaluate the monthly forecasts, each of the 51 ensemble members has been taken as input to the energy consumption and production model. For verifying the accuracy, the mean of these 51 forecasts has been calculated and verified with the Nash–Sutcliffe efficiency coefficient, whereas, for the CRPS, the averaged quantiles are used (see [73]). The resulting total predictive uncertainty comprises the uncertainty of the ML model and the NWP based forecast uncertainty, which increases with the lead-time. Since ensemble weather prediction systems are known to show bias and dispersion errors different calibration methods have been developed. Here, the NGR technique [11] is applied, which yields probabilistic forecasts that take the form of Gaussian predictive probability density functions (pdfs). The NGR predictive mean is a bias-corrected weighted average of the individual forecast models, with coefficients that can be interpreted in terms of the relative contributions of the used models. The NGR predictive variance is a linear function of the forecast variance. In this study, the NGR is applied in order to optimally combine the results of the different ML models (MLR, MARS, QRNN, QRF, KQR, GBM). Each model is run with 51 ensemble members stemming from the ENS forecast system. Thus, the overall variance will be the sum of the intra model variance and the inner model variance, which will be estimated for each lead time separately given the previous forecasts. The difference between the intra model and the overall variance can be seen in the example consumption forecast in Figure 6 on the left. In this example, the mean of four different models is shown plus/minus 3 times the standard deviation (approximately the 99.7% interval) for the four models (in light blue). In grey, the 99.7% interval is indicated after adding to the intra model variance the variance of the 51 ensemble members for each model. In Figure 6 on the right, the NGR results for this example are shown demonstrating the increased accuracy and the reduction of the uncertainty. It is also interesting to see how the difference between the intra and overall variance almost vanishes in this example because of the optimization of the NGR parameters for each lead-time separately. However, it should be mentioned that this example is a forecast from summer 2018 with a stable high pressure area over Europe lasting for a long period. This is reflected by a low increase in the ensemble spread with the lead-time.

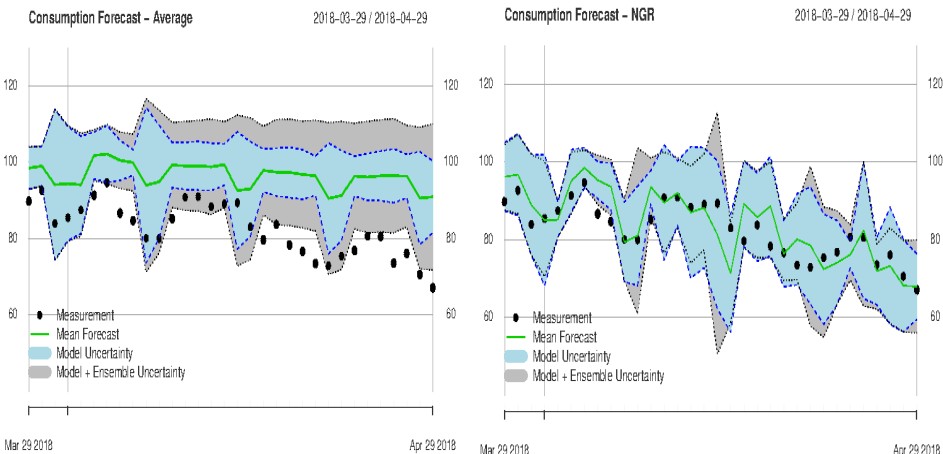

**Figure 6.** Example of a consumption forecast with simple averaging (**left side**) and the NGR approach (**right side**). The intra model uncertainty is in light blue and the overall uncertainty in grey is shown (mean plus/minus three times the standard deviation, i.e., the 99.7% interval).

Alternatively, the GBM has been tested for the combination of the various ML methods also. However, because of the limited amount of monthly forecasts available (only starting from Spring 2018 with a total of about 20 weeks of monthly forecasts issued once or twice per week), such data intense methods could not be used successfully. Nonetheless, in future studies, such ML methods for the combination will be tested thoroughly having longer time-series of forecast data at disposal. Despite the sample size of the available ENS forecast, the importance of the model combination can be

seen quite clear indicated in a much lower CRPS value for the consumption and the production model (Figures 7 and 8). For clarity and better of reading purposes, only the results for the MLR, MARS, QRF and the NGR are shown.

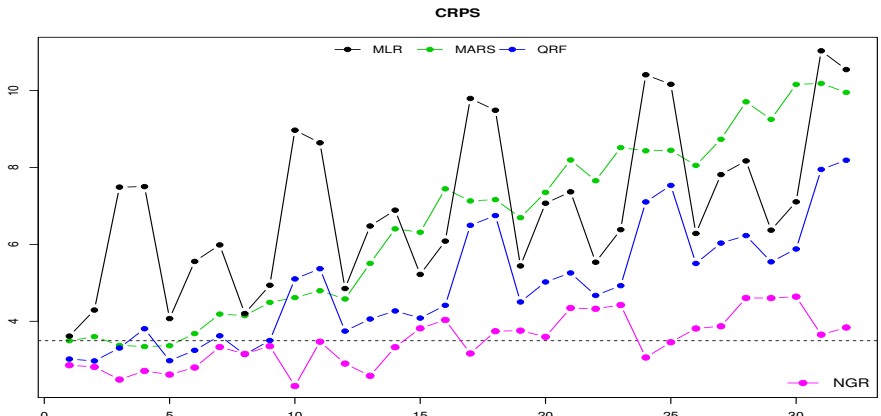

**Figure 7.** CRPS of the monthly forecasts (with daily resolution) for the consumption with a forecast horizon of 1 to 32 days (*x*-axis). For reasons of readability, only three methods (MLR—black, MARS—green, QRF—blue) and the NGR results (in magenta) are shown. Since the CRPS is negatively oriented (i.e., the lower the CRPS value, the better), the NGR shows the best performance for all lead times.

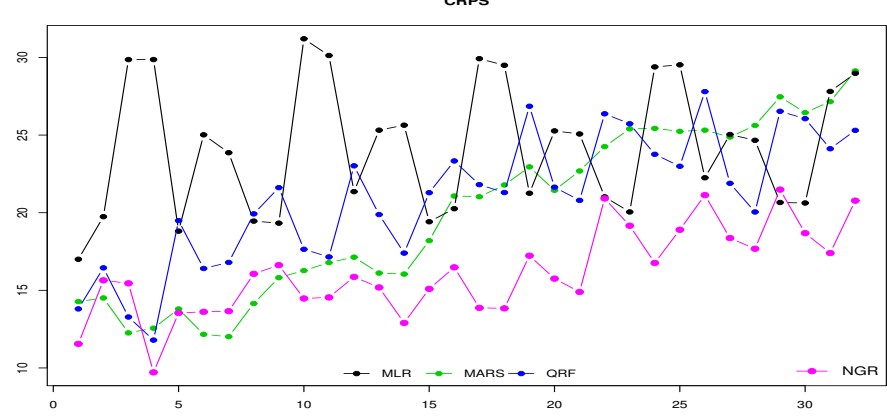

**Figure 8.** CRPS of the monthly forecasts (with daily resolution) for the production with a forecast horizon of 1 to 32 days (*x*-axis). For reasons of readability, only three methods (MLR—black, MARS—green, QRF—blue) and the NGR results (in magenta) are shown. Since the CRPS is negatively oriented (i.e., the lower the CRPS value, the better), the NGR shows the best performance for all lead times.

Since the CRPS is averaged over all forecasts, a more detailed look at examples of a single monthly consumption forecast for the middle of August 2018 is shown in Figure 9 for the MLR, MARS, QRF and the NGR model, resp. of a single monthly production forecast for the same period in Figure 10. In both figures, the 50% (in light blue) and the 99% (in grey) prediction intervals are indicated, which are much wider for the production forecasts and results in greater CRPS values. These examples also help to explain some additional properties of the resulting CRPS shown in Figures 7 and 8. For the consumption and the production, the forecasts based on the MLR are quite linear and the weekly cycle is not reproduced properly. This lacking periodicity and big uncertainty (especially for the production) is reflected in the CRPS as well, showing peaks of very high CRPS values (i.e., bad forecasts) every weekend. This periodicity is much better captured with the MARS and the

QRF model. However, both models show a uniform behaviour of the uncertainty. Only after combining the different predictions is the variability of the forecast improved, showing lower uncertainty in the beginning of the forecast period, which increases with lead time and thus reproduces better the meteorological uncertainty depending on the lead time.

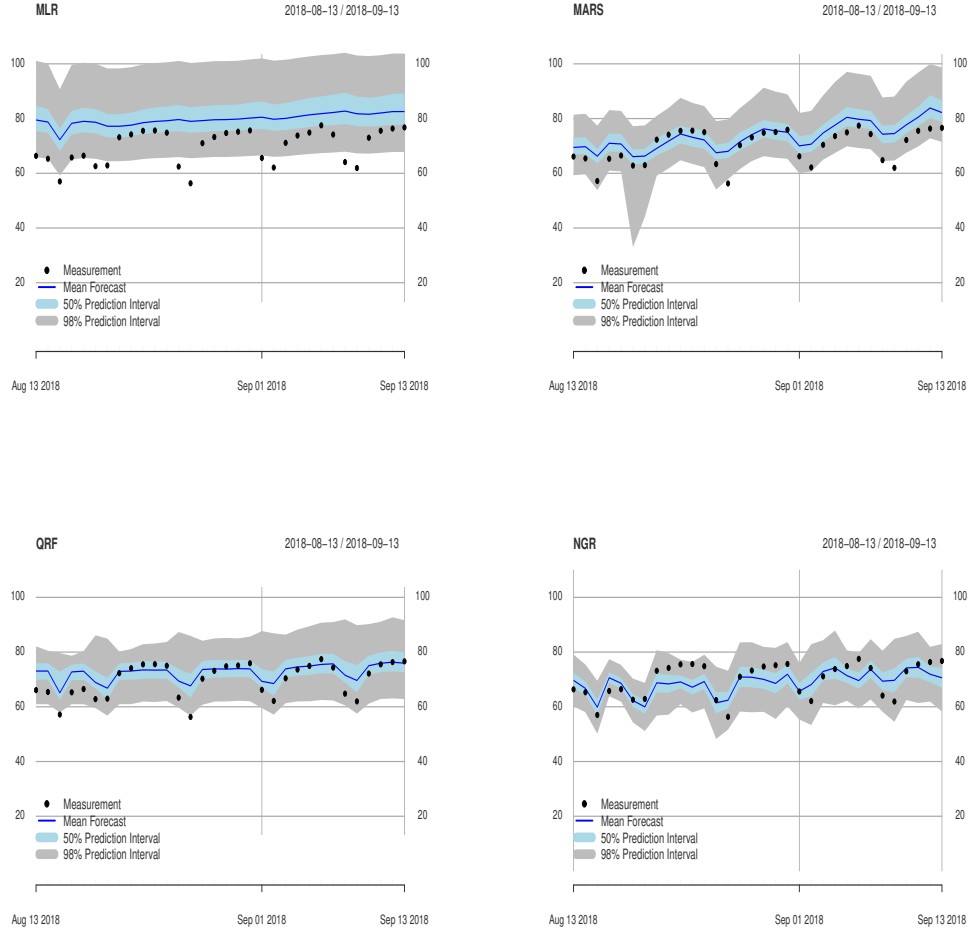

**Figure 9.** Example of monthly consumption forecasts of the MLR, MARS, QRF model and the NGR combination from the middle of August 2018 (in light blue, the 50% and in grey the 99% prediction intervals are shown).

Decision makers and energy producers could benefit from the outcome of this study in several ways:

- The usage of hydro-meteorological data, even with low spatial resolution and high uncertainty, in combination with ML methods, will significantly improve the predictability of the energy consumption/production
- The time-scale decomposition of the most important variables (temperature, resp. runoff) enhances the quality of the predictions
- Monthly weather forecasts produce skillful energy forecasts and could be used gainfully for long-term planning (e.g., changes of the hydro-power management according to forecasts of dry summer periods and taking into consideration a potential increase of the PV production)

- The estimation and the verification of the predictive uncertainty lead to more reliable predictions and forecasts, which allow end-users to evaluate potential risks and losses, having more trustworthy information available
- The application of various models and ensembles and their optimal combination reduces biases and improves the overall forecast quality and reliability
- Since hydro-meteorological data are the most important drivers of the forecast models and are often publicly available, the proposed methods could be easily transferred to different locations, catchment or regions

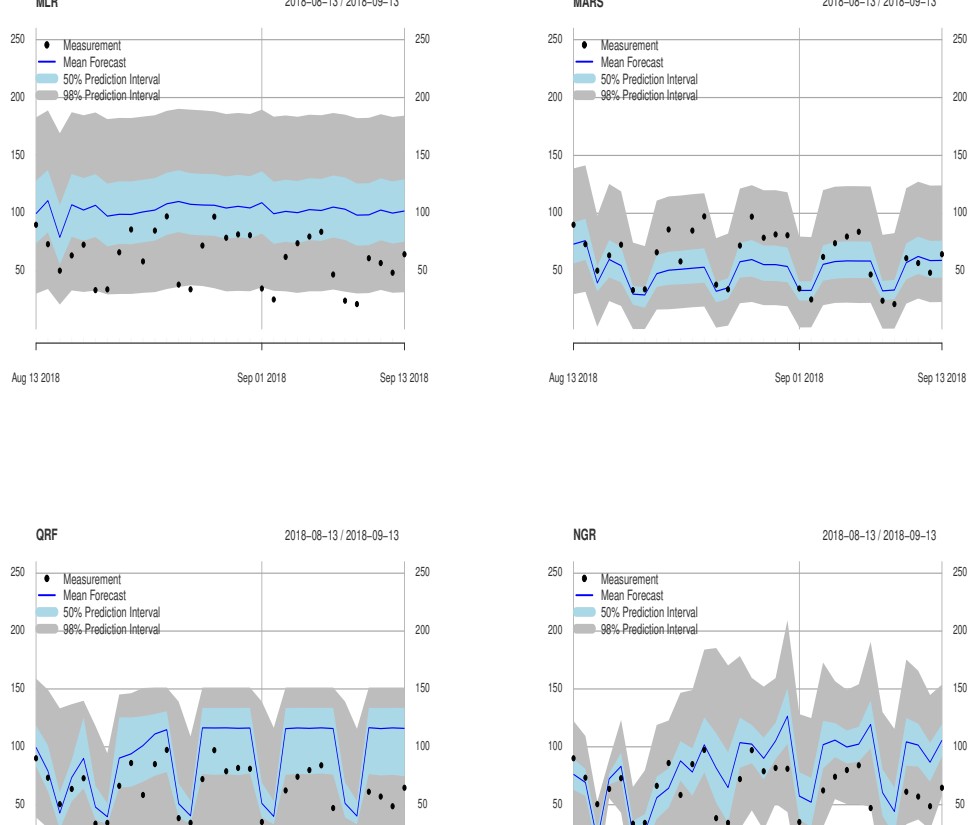

**Figure 10.** Example of monthly production forecasts of the MLR, MARS, QRF model and the NGR combination from mid of August 2018 (in light blue the 50% and in grey the 99% prediction intervals are shown).

## 4. Conclusions and Outlook

The accurate and reliable prediction of the energy consumption and production for the coming days and weeks are important for many ecological and economic aspects. In this study, the prediction of energy consumption and production is based on variables, which are available from meteorological data providers. Prediction processes that possess nonlinear properties are very attractive for Machine Learning methods. Thus, several ML methods have been compared with an emphasis on the estimated predictive uncertainties, highly relevant information for the end-users to aid with effective decision-making, which is most often neglected in the ML applications. Therefore, not only the accuracy of the predictions has been verified by the Coefficient of Determination (resp. the Nash

Sutcliffe Efficiency Measure), but also the predictive uncertainty has been evaluated by the Continuous Rank Probability Score. As benchmark models, the Multiple Linear Regression and the Quantile Regression Models have been used in order to see the possible gains in applying computational more intense models.

For the production models, the different ML methods show some significant improvements. The consumption, especially during the testing period (2018), shows quite linear behaviour, which is probably caused by the long lasting dry period from April to October in 2018. Thus, the improvements stemming from capturing nonlinearities is rather limited. Nonetheless, the ML methods show some potential to improve the skill also for this period; especially, the predictive uncertainty can be estimated sharper and more reliable in comparison to the linear models. Additionally, the first tests of applying the ML methods for forecasting the monthly consumption and production highlight the importance of applying different approaches and combining the results optimally, for example by the use of the Nonhomogeneous Gaussian Regression approach.

This study should be considered as a basis for evaluating the potentials of ML methods with very limited input and rough approximations and simplified assumptions, nonetheless with quite promising results. In particular, the proposed methods for improving the prediction and forecasting performance, like the wavelet decomposition of the main input variables (temperature and runoff), the incorporation of monthly weather forecasts, the optimal combination of different models and taking into account the predictive uncertainty in order to increase the reliability, will enhance the applicability of ML methods for the predictions of the energy consumption/production. Next, the very short-term forecasts/nowcasts (with lead-times less than 48 hours) will be coupled with an energy consumption/production model in order to evaluate the gains of applying prediction models for the next hours up to two days at very local scales. Furthermore, the incorporation of additional informations (e.g., economic predictors) and different sources of energy production (e.g., PV and wind) and their impact in the modelling chain will be tested. This will lead to much more complex methods, where probably other ML methods like Deep Learning could be more expedient.

**Author Contributions:** Writing—original draft preparation, K.B.; writing—review and editing, F.P. and M.Z.; funding acquisition, M.Z.

**Funding:** This study was conducted in the framework of the Swiss Competence Center for Energy Research—Supply of Electricity (SCCER-SoE) with funding from the Commission for Technology and Innovation CTI Grant No. 2013.0288.

**Acknowledgments:** MeteoSwiss is greatly acknowledged for providing all used meteorological data and Swissgrid for making the energy consumption/production data publicly accessible.

**Conflicts of Interest:** The authors declare no conflict of interest.

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
