# Peer review of "Machine Learning Techniques for Predicting the Energy Consumption/Production and Its Uncertainties Driven by Meteorological Observations and Forecasts"

_sustainability, doi:10.3390/su11123328_

Reviewer 1 Report

I have reviewed the Manuscript ID: sustainability-512873, with the title "Machine Learning Techniques for Predicting the Energy Consumption/ Production and its Uncertainties Driven by Meteorological Observations and Forecasts". In this paper, the authors assess the suitability of several Machine Learning (ML) methodologies for predicting the energy consumption/ production and for deriving predictive uncertainties based on the information of hydro-meteorological data. The authors conclude that the obtained results highlight the importance of optimally combining different ML methods for achieving more accurate estimates of future energy consumptions and productions with sharper prediction uncertainty estimates (i.e. narrower prediction intervals).

I consider that the paper will benefit if the authors address within the manuscript the following aspects:

Ø The "Abstract" of the paper. In the abstract the authors must also declare and briefly justify the novelty of their work in addition to the presented elements.

Ø The "Introduction" section. After declaring the novel aspects of their work, at the end of the "Introduction" section, the authors should present the structure of their paper, under the form: "The rest of the paper is structured as follows: Section 2 contains…".

Ø The "Materials and Methods" section. I consider that the authors must pay more attention to the appropriate citation of the methods and results that have been retrieved from the scientific literature. When the authors present the information in the "Materials and Methods" section, they must assume clearly their original contribution by specifying this fact and by highlighting the fact that starting from a certain point there are depicted the original and novel aspects of their research. In order to help the readers better understand the methodology of the conducted study, in the "Materials and Methods" section the authors should devise a flowchart that depicts the steps that they have processed in developing their research and most important of all, the final target. This flowchart will facilitate the understanding of the proposed approach. This diagram should be analyzed in detail within the manuscript by specifying all the elements needed for each and every step, in order to reach the final result of the study.

Ø The equations within the manuscript should be explained, demonstrated or cited, as there are some equations that have not been introduced in the literature for the first time by the authors and that are not cited.

Ø The "Results and Discussion" section. The manuscript under review will benefit if the authors improve the actual "Results and Discussion" section. In order to validate the usefulness of their research, in the "Results and Discussion" section, the authors should make a comparison between their approach from the manuscript and other similar ones that have been developed in the scientific literature for the same or related purposes. In the actual form of the manuscript there is only one cited paper in the "Results and Discussion" section and even this paper was not used in an appropriate manner in order to compare the approach from the manuscript to other ones that have been developed and used for the same purpose in the literature. I consider that the paper will benefit if the authors make a step further, beyond their analysis and provide an insight at the end of the "Results and Discussion" section regarding what they consider to be, based on the obtained results, the most important, appropriate and concrete actions that the decisional factors and all the involved parties should take in order to benefit from the results of the research conducted within the manuscript as to attain the ultimate goal of sustainability.

Ø Remarks regarding the Figures. The paper contains a lot of insufficiently explained and interpreted figures (for example, Figure 1 covers the whole page 10,  Figure 2 covers the whole page 11, Figure 3 covers the whole page 12, and all these figures are commented within the same phrase, under the form: "Based on previous studies of applicability of monthly weather forecasts for the optimization of the hydro power plant production (see [54]) the meteorological data spatially aggregated for the Verzasca catchment have been used as a surrogate for the canton Ticino (see Figure 1) and the runoff, production and consumption data are for the Canton Ticino (see Figure 2 and 3)." Similarly, page 17 contains only the Figures 6 and 7 which are commented within the same phrase, in the form: "These examples help to explain also some properties of the resulting CRPS shown in Fig. 6 and 7.", page 18 contains only Figure 8 while page 19 contains only Figure 9. The authors must explain and analyze in detail all the figures that have been inserted within the manuscript, it is not suitable to put the reader in the situation of interpreting, analyzing, continuing or refining the study from the manuscript under review.  

Ø Remarks regarding the Figures' citation. According to the Sustainability MDPI Journal's Template, all the figures should be cited in the main text as Figure 1, Figure 2, etc. In the manuscript under review, there are different ways in which this information appears (sometimes in the main text appears "Fig…", sometimes appears "Figure…") Please address this issue by modifying the way in which the figures are referred in the main text.

Ø Remarks regarding the acquired dataset. In the Section "3. Data", the authors state: "The data are provided spatially aggregated for each canton in Switzerland with 15 minutes resolution starting from the year 2015." I would like the authors to explain what is the timeframe to which the data refer. The authors must provide more details regarding the way in which they intend to solve the problems related to missing data or abnormal values if they are to occur.

Ø Remarks regarding the chosen case study. In the "Introduction" section, the authors state: "The coupling of energy models with numerical weather forecast models has been propagated by [14], but by the best knowledge of the authors no monthly ensemble prediction system has been applied so far. This will be done the first time in this paper for a region of Switzerland." The authors should highlight more the generalization capability of their approach in order to be able to justify a wider contribution that has been brought to the current state of art. In addition to this, the authors should explain in the article if (and why) only a single case study is sufficient in order to validate the generalization capability of the proposed approach. 

Ø Remarks regarding the results. In the Section "5. Conclusions and Outlook", the authors state: "Thus several ML methods have been compared with an emphasis on the estimated predictive uncertainties, a highly relevant information for the end-users to aid effective decision making, which is most often neglected in the ML applications. Therefore not only the accuracy of the predictions has been verified by the Coefficient of Determination (resp. the Nash Sutcliffe Efficiency Measure), but also the predictive uncertainty has been evaluated by the Continuous Rank Probability Score. As benchmark models the Multiple Linear Regression and the Quantile Regression Models have been used in order to see the possible gains in applying computational more intense models." In accordance to this paragraph, the authors state that the main outcome of this paper is a benchmark of pre-existing methods. I consider that these sentences should be nuanced, refined, because in their actual form, the original contribution of the current paper is not sufficiently highlighted.

Ø The "References" section is untitled.

Author Response

The authors want to thank the reviewers for their very helpful comments and their suggestions to improve the paper!

Reviewer 1:

I have reviewed the Manuscript ID: sustainability-512873, with the title "Machine Learning Techniques for Predicting the Energy Consumption/ Production and its Uncertainties Driven by Meteorological Observations and Forecasts". In this paper, the authors assess the suitability of several Machine Learning (ML) methodologies for predicting the energy consumption/ production and for deriving predictive uncertainties based on the information of hydro-meteorological data. The authors conclude that the obtained results highlight the importance of optimally combining different ML methods for achieving more accurate estimates of future energy consumptions and productions with sharper prediction uncertainty estimates (i.e. narrower prediction intervals).

I consider that the paper will benefit if the authors address within the manuscript the following aspects:

Ø The "Abstract" of the paper. In the abstract the authors must also declare and briefly justify the novelty of their work in addition to the presented elements.

In the abstract the following sentence have been included: The novelty and main focus of this study is the comparison of the capability of ML methods for producing reliable predictive uncertainties and the application of monthly weather forecasts.

Ø The "Introduction" section. After declaring the novel aspects of their work, at the end of the "Introduction" section, the authors should present the structure of their paper, under the form: "The rest of the paper is structured as follows: Section 2 contains…".

The structure is now presented as suggested (page 2, line 78-80)

Ø The "Materials and Methods" section. I consider that the authors must pay more attention to the appropriate citation of the methods and results that have been retrieved from the scientific literature. When the authors present the information in the "Materials and Methods" section, they must assume clearly their original contribution by specifying this fact and by highlighting the fact that starting from a certain point there are depicted the original and novel aspects of their research. In order to help the readers better understand the methodology of the conducted study, in the "Materials and Methods" section the authors should devise a flowchart that depicts the steps that they have processed in developing their research and most important of all, the final target. This flowchart will facilitate the understanding of the proposed approach. This diagram should be analyzed in detail within the manuscript by specifying all the elements needed for each and every step, in order to reach the final result of the study.

We would like to apologize for the missing citations and references, which have been updated now and the section has been revised thoroughly. According to the suggestion of the reviewer a flowchart has been included now (page 3) and the section has been reordered following the structure shown in the flowchart. We hope that this will help to understand the proposed approach.

Ø The equations within the manuscript should be explained, demonstrated or cited, as there are some equations that have not been introduced in the literature for the first time by the authors and that are not cited.

All the equations are referenced and cited now. However, a more detailed explanations of all the equations used by the different models would certainly go beyond the scope of this paper.

Ø The "Results and Discussion" section. The manuscript under review will benefit if the authors improve the actual "Results and Discussion" section. In order to validate the usefulness of their research, in the "Results and Discussion" section, the authors should make a comparison between their approach from the manuscript and other similar ones that have been developed in the scientific literature for the same or related purposes. In the actual form of the manuscript there is only one cited paper in the "Results and Discussion" section and even this paper was not used in an appropriate manner in order to compare the approach from the manuscript to other ones that have been developed and used for the same purpose in the literature.

In the beginning of the results section an overview of similar approaches is given and the differences are outlined indicating the novelty of our study (page 13, line  334-353).

I consider that the paper will benefit if the authors make a step further, beyond their analysis and provide an insight at the end of the "Results and Discussion" section regarding what they consider to be, based on the obtained results, the most important, appropriate and concrete actions that the decisional factors and all the involved parties should take in order to benefit from the results of the research conducted within the manuscript as to attain the ultimate goal of sustainability.

On page 17 the most important outcomes and the benefits for potential applicants are summarized (line 434 – 451)

Ø Remarks regarding the Figures. The paper contains a lot of insufficiently explained and interpreted figures (for example, Figure 1 covers the whole page 10,  Figure 2 covers the whole page 11, Figure 3 covers the whole page 12, and all these figures are commented within the same phrase, under the form: "Based on previous studies of applicability of monthly weather forecasts for the optimization of the hydro power plant production (see [54]) the meteorological data spatially aggregated for the Verzasca catchment have been used as a surrogate for the canton Ticino (see Figure 1) and the runoff, production and consumption data are for the Canton Ticino (see Figure 2 and 3)." Similarly, page 17 contains only the Figures 6 and 7 which are commented within the same phrase, in the form: "These examples help to explain also some properties of the resulting CRPS shown in Fig. 6 and 7.", page 18 contains only Figure 8 while page 19 contains only Figure 9. The authors must explain and analyze in detail all the figures that have been inserted within the manuscript, it is not suitable to put the reader in the situation of interpreting, analyzing, continuing or refining the study from the manuscript under review.  

All the figures are described in more detail now and the captions are updated.

Ø Remarks regarding the Figures' citation. According to the Sustainability MDPI Journal's Template, all the figures should be cited in the main text as Figure 1, Figure 2, etc. In the manuscript under review, there are different ways in which this information appears (sometimes in the main text appears "Fig…", sometimes appears "Figure…") Please address this issue by modifying the way in which the figures are referred in the main text.

Done

Ø Remarks regarding the acquired dataset. In the Section "3. Data", the authors state: "The data are provided spatially aggregated for each canton in Switzerland with 15 minutes resolution starting from the year 2015." I would like the authors to explain what is the timeframe to which the data refer. The authors must provide more details regarding the way in which they intend to solve the problems related to missing data or abnormal values if they are to occur.

The time frame has been included (page 4, line 104) and the issue of missing data has been mentioned (line 120)

Ø Remarks regarding the chosen case study. In the "Introduction" section, the authors state: "The coupling of energy models with numerical weather forecast models has been propagated by [14], but by the best knowledge of the authors no monthly ensemble prediction system has been applied so far. This will be done the first time in this paper for a region of Switzerland." The authors should highlight more the generalization capability of their approach in order to be able to justify a wider contribution that has been brought to the current state of art. In addition to this, the authors should explain in the article if (and why) only a single case study is sufficient in order to validate the generalization capability of the proposed approach. 

We agree with the reviewer that generalization is an important aspect and we have mentioned now that our methods could be transferred easily to other locations (see page 17, line 455-457). We also would like to stress that we have referenced all the used R packages, which are publically available and allows the potential users to apply the proposed methods with their own data sets. One of the main objective of this paper was to test how these ML methods work with insufficient and imprecise data. We could demonstrate that even with limited data these methods show improvements and we are certain, that with better data sets the methods can be successfully transferred and applied to any location. However, we will test this in a next step.

Ø Remarks regarding the results. In the Section "5. Conclusions and Outlook", the authors state: "Thus several ML methods have been compared with an emphasis on the estimated predictive uncertainties, a highly relevant information for the end-users to aid effective decision making, which is most often neglected in the ML applications. Therefore not only the accuracy of the predictions has been verified by the Coefficient of Determination (resp. the Nash Sutcliffe Efficiency Measure), but also the predictive uncertainty has been evaluated by the Continuous Rank Probability Score. As benchmark models the Multiple Linear Regression and the Quantile Regression Models have been used in order to see the possible gains in applying computational more intense models." In accordance to this paragraph, the authors state that the main outcome of this paper is a benchmark of pre-existing methods. I consider that these sentences should be nuanced, refined, because in their actual form, the original contribution of the current paper is not sufficiently highlighted.

We have included sentences in the conclusion section to highlight the contribution (page 20, line 483-487).

Ø The "References" section is untitled.

Done

Reviewer 2 Report

The paper uses several machine learning techniques to forecast energy consumption and production using meteorological forecasts.

The novelty of the paper is explained in the introduction and literature review is adequate.

However, my main concern with the paper is that it is not engaging and is difficult to follow. The practical aspects of the methodology and results are not shown properly, relying mainly on statistical numbers. My impression is that at its current form, it will not attract readers and will remain largely unnoticed.

I find also difficult to understand what exactly the paper forecasts. It seems that only monthly aggregated forecasts are made (judging from Figs 2 and 3), although data are 15 min time-series. I could understand the importance of having next day or next hour predictions at 15 min steps, but for monthly averages i am not sure of their practical value. Maybe though i have not well understood the method, and this goes back to my previous comment.

I am afraid i do not have any other specific comment to make to help authors improve their paper. I would advise to clarify the above points and try to highlight the practical aspects of their work.

Author Response

The authors want to thank the reviewers for their very helpful comments and their suggestions to improve the paper!

The paper uses several machine learning techniques to forecast energy consumption and production using meteorological forecasts.

The novelty of the paper is explained in the introduction and literature review is adequate.

However, my main concern with the paper is that it is not engaging and is difficult to follow. The practical aspects of the methodology and results are not shown properly, relying mainly on statistical numbers. My impression is that at its current form, it will not attract readers and will remain largely unnoticed.

We have rearranged the paper structure  and included a flowchart (see Reviewer 1) to improve the readability. We have also stressed now at several places (in the sections Methods, Results and Conclusions) the importance of the proposed methods for potential end-users and hope that this will help to attract readers now.

I find also difficult to understand what exactly the paper forecasts. It seems that only monthly aggregated forecasts are made (judging from Figs 2 and 3), although data are 15 min time-series. I could understand the importance of having next day or next hour predictions at 15 min steps, but for monthly averages i am not sure of their practical value. Maybe though i have not well understood the method, and this goes back to my previous comment.

As described in the Data section the data used are daily aggregates. We have also discussed several times with potential end-users (hydro power producers, energy providers), who confirmed the importance of having reliable forecasts for the upcoming weeks. These discussions encouraged us to test the methods for monthly time scales. 

I am afraid i do not have any other specific comment to make to help authors improve their paper. I would advise to clarify the above points and try to highlight the practical aspects of their work.

Round  2

Reviewer 1 Report

I have reviewed the revised version of the Manuscript ID: sustainability-512873, with the title "Machine Learning Techniques for Predicting the Energy Consumption/Production and its Uncertainties Driven by Meteorological Observations and Forecasts" and I can conclude that the authors have addressed the most important signaled issues, therefore improving the manuscript.